# Neural Word Embedding
# as Implicit Matrix Factorization

**Omer Levy**
Department of Computer Science
Bar-Ilan University
omerlevy@gmail.com

**Yoav Goldberg**
Department of Computer Science
Bar-Ilan University
yoav.goldberg@gmail.com

## Abstract

We analyze skip-gram with negative-sampling (SGNS), a word embedding method introduced by Mikolov et al., and show that it is implicitly factorizing a word-context matrix, whose cells are the pointwise mutual information (PMI) of the respective word and context pairs, shifted by a global constant. We find that another embedding method, NCE, is implicitly factorizing a similar matrix, where each cell is the (shifted) log conditional probability of a word given its context. We show that using a sparse *Shifted Positive PMI* word-context matrix to represent words improves results on two word similarity tasks and one of two analogy tasks. When dense low-dimensional vectors are preferred, exact factorization with SVD can achieve solutions that are at least as good as SGNS's solutions for word similarity tasks. On analogy questions SGNS remains superior to SVD. We conjecture that this stems from the weighted nature of SGNS's factorization.

## 1 Introduction

Most tasks in natural language processing and understanding involve looking at words, and could benefit from word representations that do not treat individual words as unique symbols, but instead reflect similarities and dissimilarities between them. The common paradigm for deriving such representations is based on the distributional hypothesis of Harris [15], which states that words in similar contexts have similar meanings. This has given rise to many word representation methods in the NLP literature, the vast majority of whom can be described in terms of a word-context matrix $M$ in which each row $i$ corresponds to a word, each column $j$ to a context in which the word appeared, and each matrix entry $M_{ij}$ corresponds to some association measure between the word and the context. Words are then represented as rows in $M$ or in a dimensionality-reduced matrix based on $M$.

Recently, there has been a surge of work proposing to represent words as dense vectors, derived using various training methods inspired from neural-network language modeling [3, 9, 23, 21]. These representations, referred to as "neural embeddings" or "word embeddings", have been shown to perform well in a variety of NLP tasks [26, 10, 1]. In particular, a sequence of papers by Mikolov and colleagues [20, 21] culminated in the skip-gram with negative-sampling (SGNS) training method which is both efficient to train and provides state-of-the-art results on various linguistic tasks. The training method (as implemented in the `word2vec` software package) is highly popular, but not well understood. While it is clear that the training objective follows the distributional hypothesis – by trying to maximize the dot-product between the vectors of frequently occurring word-context pairs, and minimize it for random word-context pairs – very little is known about the quantity being optimized by the algorithm, or the reason it is expected to produce good word representations.

In this work, we aim to broaden the theoretical understanding of neural-inspired word embeddings. Specifically, we cast SGNS's training method as weighted matrix factorization, and show that its objective is implicitly factorizing a *shifted PMI matrix* – the well-known word-context PMI matrix from the word-similarity literature, shifted by a constant offset. A similar result holds also for the

NCE embedding method of Mnih and Kavukcuoglu [24]. While it is impractical to directly use the very high-dimensional and dense shifted PMI matrix, we propose to approximate it with the *positive* shifted PMI matrix (Shifted PPMI), which is sparse. Shifted PPMI is far better at optimizing SGNS's objective, and performs slightly better than `word2vec` derived vectors on several linguistic tasks.

Finally, we suggest a simple spectral algorithm that is based on performing SVD over the Shifted PPMI matrix. The spectral algorithm outperforms both SGNS and the Shifted PPMI matrix on the word similarity tasks, and is scalable to large corpora. However, it lags behind the SGNS-derived representation on word-analogy tasks. We conjecture that this behavior is related to the fact that SGNS performs *weighted* matrix factorization, giving more influence to frequent pairs, as opposed to SVD, which gives the same weight to all matrix cells. While the weighted and non-weighted objectives share the same optimal solution (perfect reconstruction of the shifted PMI matrix), they result in different generalizations when combined with dimensionality constraints.

## 2 Background: Skip-Gram with Negative Sampling (SGNS)

Our departure point is SGNS – the skip-gram neural embedding model introduced in [20] trained using the negative-sampling procedure presented in [21]. In what follows, we summarize the SGNS model and introduce our notation. A detailed derivation of the SGNS model is available in [14].

**Setting and Notation** The skip-gram model assumes a corpus of words $w \in V_W$ and their contexts $c \in V_C$, where $V_W$ and $V_C$ are the word and context vocabularies. In [20, 21] the words come from an unannotated textual corpus of words $w_1, w_2, \ldots, w_n$ (typically $n$ is in the billions) and the contexts for word $w_i$ are the words surrounding it in an $L$-sized window $w_{i-L}, \ldots, w_{i-1}, w_{i+1}, \ldots, w_{i+L}$. Other definitions of contexts are possible [18]. We denote the collection of observed words and context pairs as $D$. We use $\#(w, c)$ to denote the number of times the pair $(w, c)$ appears in $D$. Similarly, $\#(w) = \sum_{c' \in V_C} \#(w, c')$ and $\#(c) = \sum_{w' \in V_W} \#(w', c)$ are the number of times $w$ and $c$ occurred in $D$, respectively.

Each word $w \in V_W$ is associated with a vector $\vec{w} \in \mathbb{R}^d$ and similarly each context $c \in V_C$ is represented as a vector $\vec{c} \in \mathbb{R}^d$, where $d$ is the embedding's dimensionality. The entries in the vectors are latent, and treated as parameters to be learned. We sometimes refer to the vectors $\vec{w}$ as rows in a $|V_W| \times d$ matrix $W$, and to the vectors $\vec{c}$ as rows in a $|V_C| \times d$ matrix $C$. In such cases, $W_i$ ($C_i$) refers to the vector representation of the $i$th word (context) in the corresponding vocabulary. When referring to embeddings produced by a specific method $x$, we will usually use $W^x$ and $C^x$ explicitly, but may use just $W$ and $C$ when the method is clear from the discussion.

**SGNS's Objective** Consider a word-context pair $(w, c)$. Did this pair come from the observed data $D$? Let $P(D = 1|w, c)$ be the probability that $(w, c)$ came from the data, and $P(D = 0|w, c) = 1 - P(D = 1|w, c)$ the probability that $(w, c)$ did not. The distribution is modeled as:

$$P(D = 1|w, c) = \sigma(\vec{w} \cdot \vec{c}) = \frac{1}{1 + e^{-\vec{w} \cdot \vec{c}}}$$

where $\vec{w}$ and $\vec{c}$ (each a $d$-dimensional vector) are the model parameters to be learned.

The negative sampling objective tries to maximize $P(D = 1|w, c)$ for observed $(w, c)$ pairs while maximizing $P(D = 0|w, c)$ for randomly sampled "negative" examples (hence the name "negative sampling"), under the assumption that randomly selecting a context for a given word is likely to result in an unobserved $(w, c)$ pair. SGNS's objective for a single $(w, c)$ observation is then:

$$\log \sigma(\vec{w} \cdot \vec{c}) + k \cdot \mathbb{E}_{c_N \sim P_D} \left[ \log \sigma(-\vec{w} \cdot \vec{c}_N) \right] \tag{1}$$

where $k$ is the number of "negative" samples and $c_N$ is the sampled context, drawn according to the empirical unigram distribution $P_D(c) = \frac{\#(c)}{|D|}$. [1]

The objective is trained in an online fashion using stochastic gradient updates over the observed pairs in the corpus $D$. The global objective then sums over the observed $(w, c)$ pairs in the corpus:

$$\ell = \sum_{w \in V_W} \sum_{c \in V_C} \#(w, c) \left( \log \sigma(\vec{w} \cdot \vec{c}) + k \cdot \mathbb{E}_{c_N \sim P_D} \left[ \log \sigma(-\vec{w} \cdot \vec{c}_N) \right] \right) \qquad (2)$$

Optimizing this objective makes observed word-context pairs have similar embeddings, while scattering unobserved pairs. Intuitively, words that appear in similar contexts should have similar embeddings, though we are not familiar with a formal proof that SGNS does indeed maximize the dot-product of similar words.

## 3   SGNS as Implicit Matrix Factorization

SGNS embeds both words and their contexts into a low-dimensional space $\mathbb{R}^d$, resulting in the word and context matrices $W$ and $C$. The rows of matrix $W$ are typically used in NLP tasks (such as computing word similarities) while $C$ is ignored. It is nonetheless instructive to consider the product $W \cdot C^\top = M$. Viewed this way, SGNS can be described as *factorizing* an implicit matrix $M$ of dimensions $|V_W| \times |V_C|$ into two smaller matrices.

Which matrix is being factorized? A matrix entry $M_{ij}$ corresponds to the dot product $W_i \cdot C_j = \vec{w}_i \cdot \vec{c}_j$. Thus, SGNS is factorizing a matrix in which each row corresponds to a word $w \in V_W$, each column corresponds to a context $c \in V_C$, and each cell contains a quantity $f(w, c)$ reflecting the strength of association between that particular word-context pair. Such word-context association matrices are very common in the NLP and word-similarity literature, see e.g. [29, 2]. That said, the objective of SGNS (equation 2) does not explicitly state what this association metric is. What can we say about the association function $f(w, c)$? In other words, which matrix is SGNS factorizing?

### 3.1   Characterizing the Implicit Matrix

Consider the global objective (equation 2) above. For sufficiently large dimensionality $d$ (i.e. allowing for a perfect reconstruction of $M$), each product $\vec{w} \cdot \vec{c}$ can assume a value independently of the others. Under these conditions, we can treat the objective $\ell$ as a function of independent $\vec{w} \cdot \vec{c}$ terms, and find the values of these terms that maximize it.

We begin by rewriting equation 2:

$$\ell = \sum_{w \in V_W} \sum_{c \in V_C} \#(w, c) \left( \log \sigma(\vec{w} \cdot \vec{c}) \right) + \sum_{w \in V_W} \sum_{c \in V_C} \#(w, c) \left( k \cdot \mathbb{E}_{c_N \sim P_D} \left[ \log \sigma(-\vec{w} \cdot \vec{c}_N) \right] \right)$$

$$= \sum_{w \in V_W} \sum_{c \in V_C} \#(w, c) \left( \log \sigma(\vec{w} \cdot \vec{c}) \right) + \sum_{w \in V_W} \#(w) \left( k \cdot \mathbb{E}_{c_N \sim P_D} \left[ \log \sigma(-\vec{w} \cdot \vec{c}_N) \right] \right) \qquad (3)$$

and explicitly expressing the expectation term:

$$\mathbb{E}_{c_N \sim P_D} \left[ \log \sigma(-\vec{w} \cdot \vec{c}_N) \right] = \sum_{c_N \in V_C} \frac{\#(c_N)}{|D|} \log \sigma(-\vec{w} \cdot \vec{c}_N)$$

$$= \frac{\#(c)}{|D|} \log \sigma(-\vec{w} \cdot \vec{c}) + \sum_{c_N \in V_C \setminus \{c\}} \frac{\#(c_N)}{|D|} \log \sigma(-\vec{w} \cdot \vec{c}_N) \qquad (4)$$

Combining equations 3 and 4 reveals the local objective for a *specific* $(w, c)$ pair:

$$\ell(w, c) = \#(w, c) \log \sigma(\vec{w} \cdot \vec{c}) + k \cdot \#(w) \cdot \frac{\#(c)}{|D|} \log \sigma(-\vec{w} \cdot \vec{c}) \qquad (5)$$

To optimize the objective, we define $x = \vec{w} \cdot \vec{c}$ and find its partial derivative with respect to $x$:

$$\frac{\partial \ell}{\partial x} = \#(w, c) \cdot \sigma(-x) - k \cdot \#(w) \cdot \frac{\#(c)}{|D|} \cdot \sigma(x)$$

We compare the derivative to zero, and after some simplification, arrive at:

$$e^{2x} - \left( \frac{\#(w, c)}{k \cdot \#(w) \cdot \frac{\#(c)}{|D|}} - 1 \right) e^x - \frac{\#(w, c)}{k \cdot \#(w) \cdot \frac{\#(c)}{|D|}} = 0$$

If we define $y = e^x$, this equation becomes a quadratic equation of $y$, which has two solutions, $y = -1$ (which is invalid given the definition of $y$) and:

$$y = \frac{\#(w,c)}{k \cdot \#(w) \cdot \frac{\#(c)}{|D|}} = \frac{\#(w,c) \cdot |D|}{\#w \cdot \#(c)} \cdot \frac{1}{k}$$

Substituting $y$ with $e^x$ and $x$ with $\vec{w} \cdot \vec{c}$ reveals:

$$\vec{w} \cdot \vec{c} = \log\left(\frac{\#(w,c) \cdot |D|}{\#(w) \cdot \#(c)} \cdot \frac{1}{k}\right) = \log\left(\frac{\#(w,c) \cdot |D|}{\#(w) \cdot \#(c)}\right) - \log k \tag{6}$$

Interestingly, the expression $\log\left(\frac{\#(w,c) \cdot |D|}{\#(w) \cdot \#(c)}\right)$ is the well-known pointwise mutual information (PMI) of $(w,c)$, which we discuss in depth below.

Finally, we can describe the matrix $M$ that SGNS is factorizing:

$$M_{ij}^{\text{SGNS}} = W_i \cdot C_j = \vec{w}_i \cdot \vec{c}_j = PMI(w_i, c_j) - \log k \tag{7}$$

For a negative-sampling value of $k = 1$, the SGNS objective is factorizing a word-context matrix in which the association between a word and its context is measured by $f(w,c) = PMI(w,c)$. We refer to this matrix as the *PMI matrix*, $M^{PMI}$. For negative-sampling values $k > 1$, SGNS is factorizing a *shifted PMI matrix* $M^{PMI_k} = M^{PMI} - \log k$.

Other embedding methods can also be cast as factorizing implicit word-context matrices. Using a similar derivation, it can be shown that noise-contrastive estimation (NCE) [24] is factorizing the (shifted) log-conditional-probability matrix:

$$M_{ij}^{\text{NCE}} = \vec{w}_i \cdot \vec{c}_j = \log\left(\frac{\#(w,c)}{\#(c)}\right) - \log k = \log P(w|c) - \log k \tag{8}$$

## 3.2 Weighted Matrix Factorization

We obtained that SGNS's objective is optimized by setting $\vec{w} \cdot \vec{c} = PMI(w,c) - \log k$ for every $(w,c)$ pair. However, this assumes that the dimensionality of $\vec{w}$ and $\vec{c}$ is high enough to allow for perfect reconstruction. When perfect reconstruction is not possible, some $\vec{w} \cdot \vec{c}$ products must deviate from their optimal values. Looking at the pair-specific objective (equation 5) reveals that the loss for a pair $(w,c)$ depends on its number of observations ($\#(w,c)$) and expected negative samples ($k \cdot \#(w) \cdot \#(c)/|D|$). SGNS's objective can now be cast as a *weighted matrix factorization* problem, seeking the optimal $d$-dimensional factorization of the matrix $M^{PMI} - \log k$ under a metric which pays more for deviations on frequent $(w,c)$ pairs than deviations on infrequent ones.

## 3.3 Pointwise Mutual Information

Pointwise mutual information is an information-theoretic association measure between a pair of discrete outcomes $x$ and $y$, defined as:

$$PMI(x,y) = \log \frac{P(x,y)}{P(x)P(y)} \tag{9}$$

In our case, $PMI(w,c)$ measures the association between a word $w$ and a context $c$ by calculating the log of the ratio between their joint probability (the frequency in which they occur together) and their marginal probabilities (the frequency in which they occur independently). PMI can be estimated empirically by considering the actual number of observations in a corpus:

$$PMI(w,c) = \log \frac{\#(w,c) \cdot |D|}{\#(w) \cdot \#(c)} \tag{10}$$

The use of PMI as a measure of association in NLP was introduced by Church and Hanks [8] and widely adopted for word similarity tasks [11, 27, 29].

Working with the PMI matrix presents some computational challenges. The rows of $M^{\text{PMI}}$ contain many entries of word-context pairs $(w,c)$ that were never observed in the corpus, for which

$PMI(w, c) = \log 0 = -\infty$. Not only is the matrix ill-defined, it is also dense, which is a major practical issue because of its huge dimensions $|V_W| \times |V_C|$. One could smooth the probabilities using, for instance, a Dirichlet prior by adding a small "fake" count to the underlying counts matrix, rendering all word-context pairs observed. While the resulting matrix will not contain any infinite values, it will remain dense.

An alternative approach, commonly used in NLP, is to replace the $M^{\text{PMI}}$ matrix with $M_0^{\text{PMI}}$, in which $PMI(w, c) = 0$ in cases $\#(w, c) = 0$, resulting in a sparse matrix. We note that $M_0^{\text{PMI}}$ is inconsistent, in the sense that observed but "bad" (uncorrelated) word-context pairs have a negative matrix entry, while unobserved (hence worse) ones have 0 in their corresponding cell. Consider for example a pair of relatively frequent words (high $P(w)$ and $P(c)$) that occur only once together. There is strong evidence that the words tend not to appear together, resulting in a negative PMI value, and hence a negative matrix entry. On the other hand, a pair of frequent words (same $P(w)$ and $P(c)$) that is *never* observed occurring together in the corpus, will receive a value of 0.

A sparse and consistent alternative from the NLP literature is to use the *positive PMI* (PPMI) metric, in which all negative values are replaced by 0:

$$PPMI(w, c) = \max(PMI(w, c), 0) \tag{11}$$

When representing words, there is some intuition behind ignoring negative values: humans can easily think of *positive* associations (e.g. "Canada" and "snow") but find it much harder to invent *negative* ones ("Canada" and "desert"). This suggests that the perceived similarity of two words is more influenced by the positive context they share than by the negative context they share. It therefore makes some intuitive sense to discard the negatively associated contexts and mark them as "uninformative" (0) instead.[2] Indeed, it was shown that the PPMI metric performs very well on semantic similarity tasks [5].

Both $M_0^{\text{PMI}}$ and $M^{\text{PPMI}}$ are well known to the NLP community. In particular, systematic comparisons of various word-context association metrics show that PMI, and more so PPMI, provide the best results for a wide range of word-similarity tasks [5, 16]. It is thus interesting that the PMI matrix emerges as the optimal solution for SGNS's objective.

# 4   Alternative Word Representations

As SGNS with $k = 1$ is attempting to implicitly factorize the familiar matrix $M^{\text{PMI}}$, a natural algorithm would be to use the rows of $M^{\text{PPMI}}$ *directly* when calculating word similarities. Though PPMI is only an approximation of the original PMI matrix, it still brings the objective function very close to its optimum (see Section 5.1). In this section, we propose two alternative word representations that build upon $M^{\text{PPMI}}$.

## 4.1   Shifted PPMI

While the PMI matrix emerges from SGNS with $k = 1$, it was shown that different values of $k$ can substantially improve the resulting embedding. With $k > 1$, the association metric in the implicitly factorized matrix is $PMI(w, c) - \log(k)$. This suggests the use of *Shifted* PPMI (SPPMI), a novel association metric which, to the best of our knowledge, was not explored in the NLP and word-similarity communities:

$$SPPMI_k(w, c) = \max(PMI(w, c) - \log k, 0) \tag{12}$$

As with SGNS, certain values of $k$ can improve the performance of $M^{\text{SPPMI}_k}$ on different tasks.

## 4.2   Spectral Dimensionality Reduction: SVD over Shifted PPMI

While sparse vector representations work well, there are also advantages to working with dense low-dimensional vectors, such as improved computational efficiency and, arguably, better generalization.

An alternative matrix factorization method to SGNS's stochastic gradient training is truncated Singular Value Decomposition (SVD) – a basic algorithm from linear algebra which is used to achieve the optimal rank $d$ factorization with respect to $L_2$ loss [12]. SVD factorizes $M$ into the product of three matrices $U \cdot \Sigma \cdot V^\top$, where $U$ and $V$ are orthonormal and $\Sigma$ is a diagonal matrix of singular values. Let $\Sigma_d$ be the diagonal matrix formed from the top $d$ singular values, and let $U_d$ and $V_d$ be the matrices produced by selecting the corresponding columns from $U$ and $V$. The matrix $M_d = U_d \cdot \Sigma_d \cdot V_d^\top$ is the matrix of rank $d$ that best approximates the original matrix $M$, in the sense that it minimizes the approximation errors. That is, $M_d = \arg\min_{Rank(M')=d} \|M' - M\|_2$.

When using SVD, the dot-products between the rows of $W = U_d \cdot \Sigma_d$ are equal to the dot-products between rows of $M_d$. In the context of word-context matrices, the dense, $d$ dimensional rows of $W$ are perfect substitutes for the very high-dimensional rows of $M_d$. Indeed another common approach in the NLP literature is factorizing the PPMI matrix $M^{\text{PPMI}}$ with SVD, and then taking the rows of $W^{\text{SVD}} = U_d \cdot \Sigma_d$ and $C^{\text{SVD}} = V_d$ as word and context representations, respectively. However, using the rows of $W^{\text{SVD}}$ as word representations consistently under-perform the $W^{\text{SGNS}}$ embeddings derived from SGNS when evaluated on semantic tasks.

**Symmetric SVD**  We note that in the SVD-based factorization, the resulting word and context matrices have very different properties. In particular, the context matrix $C^{\text{SVD}}$ is orthonormal while the word matrix $W^{\text{SVD}}$ is not. On the other hand, the factorization achieved by SGNS's training procedure is much more "symmetric", in the sense that neither $W^{\text{W2V}}$ nor $C^{\text{W2V}}$ is orthonormal, and no particular bias is given to either of the matrices in the training objective. We therefore propose achieving similar symmetry with the following factorization:

$$W^{\text{SVD}_{1/2}} = U_d \cdot \sqrt{\Sigma_d} \qquad C^{\text{SVD}_{1/2}} = V_d \cdot \sqrt{\Sigma_d} \tag{13}$$

While it is not theoretically clear why the symmetric approach is better for semantic tasks, it does work much better empirically.[3]

**SVD versus SGNS**  The spectral algorithm has two computational advantages over stochastic gradient training. First, it is exact, and does not require learning rates or hyper-parameter tuning. Second, it can be easily trained on count-aggregated data (i.e. $\{(w, c, \#(w, c))\}$ triplets), making it applicable to much larger corpora than SGNS's training procedure, which requires each observation of $(w, c)$ to be presented separately.

On the other hand, the stochastic gradient method has advantages as well: in contrast to SVD, it distinguishes between observed and unobserved events; SVD is known to suffer from unobserved values [17], which are very common in word-context matrices. More importantly, SGNS's objective weighs different $(w, c)$ pairs differently, preferring to assign correct values to frequent $(w, c)$ pairs while allowing more error for infrequent pairs (see Section 3.2). Unfortunately, exact weighted SVD is a hard computational problem [25]. Finally, because SGNS cares only about observed (and sampled) $(w, c)$ pairs, it does not require the underlying matrix to be a sparse one, enabling optimization of dense matrices, such as the exact $PMI - \log k$ matrix. The same is not feasible when using SVD.

An interesting middle-ground between SGNS and SVD is the use of stochastic matrix factorization (SMF) approaches, common in the collaborative filtering literature [17]. In contrast to SVD, the SMF approaches are not exact, and do require hyper-parameter tuning. On the other hand, they are better than SVD at handling unobserved values, and can integrate importance weighting for examples, much like SGNS's training procedure. However, like SVD and unlike SGNS's procedure, the SMF approaches work over aggregated $(w, c)$ statistics allowing $(w, c, f(w, c))$ triplets as input, making the optimization objective more direct, and scalable to significantly larger corpora. SMF approaches have additional advantages over both SGNS and SVD, such as regularization, opening the way to a range of possible improvements. We leave the exploration of SMF-based algorithms for word embeddings to future work.

| Method | PMI$-\log k$ | SPPMI | SVD | | | SGNS | | |
|---|---|---|---|---|---|---|---|---|
| | | | $d = 100$ | $d = 500$ | $d = 1000$ | $d = 100$ | $d = 500$ | $d = 1000$ |
| $k = 1$ | 0% | 0.00009% | 26.1% | 25.2% | 24.2% | 31.4% | 29.4% | 7.40% |
| $k = 5$ | 0% | 0.00004% | 95.8% | 95.1% | 94.9% | 39.3% | 36.0% | 7.13% |
| $k = 15$ | 0% | 0.00002% | 266% | 266% | 265% | 7.80% | 6.37% | 5.97% |

Table 1: Percentage of deviation from the optimal objective value (lower values are better). See 5.1 for details.

## 5 Empirical Results

We compare the matrix-based algorithms to SGNS in two aspects. First, we measure how well each algorithm optimizes the objective, and then proceed to evaluate the methods on various linguistic tasks. We find that for some tasks there is a large discrepancy between optimizing the objective and doing well on the linguistic task.

**Experimental Setup** All models were trained on English Wikipedia, pre-processed by removing non-textual elements, sentence splitting, and tokenization. The corpus contains 77.5 million sentences, spanning 1.5 billion tokens. All models were derived using a window of 2 tokens to each side of the focus word, ignoring words that appeared less than 100 times in the corpus, resulting in vocabularies of 189,533 terms for both words and contexts. To train the SGNS models, we used a modified version of `word2vec` which receives a sequence of pre-extracted word-context pairs [18].[4] We experimented with three values of $k$ (number of negative samples in SGNS, shift parameter in PMI-based methods): 1, 5, 15. For SVD, we take $W = U_d \cdot \sqrt{\Sigma_d}$ as explained in Section 4.

### 5.1 Optimizing the Objective

Now that we have an analytical solution for the objective, we can measure how well each algorithm optimizes this objective in practice. To do so, we calculated $\ell$, the value of the objective (equation 2) given each word (and context) representation.[5] For sparse matrix representations, we substituted $\vec{w} \cdot \vec{c}$ with the matching cell's value (e.g. for SPPMI, we set $\vec{w} \cdot \vec{c} = \max(\text{PMI}(w, c) - \log k, 0)$). Each algorithm's $\ell$ value was compared to $\ell_{Opt}$, the objective when setting $\vec{w} \cdot \vec{c} = \text{PMI}(w, c) - \log k$, which was shown to be optimal (Section 3.1). The percentage of deviation from the optimum is defined by $(\ell - \ell_{Opt})/(\ell_{Opt})$ and presented in table 1.

We observe that SPPMI is indeed a near-perfect approximation of the optimal solution, even though it discards a lot of information when considering only positive cells. We also note that for the factorization methods, increasing the dimensionality enables better solutions, as expected. SVD is slightly better than SGNS at optimizing the objective for $d \leq 500$ and $k = 1$. However, while SGNS is able to leverage higher dimensions and reduce its error significantly, SVD fails to do so. Furthermore, SVD becomes very erroneous as $k$ increases. We hypothesize that this is a result of the increasing number of zero-cells, which may cause SVD to prefer a factorization that is very close to the zero matrix, since SVD's $L_2$ objective is unweighted, and does not distinguish between observed and unobserved matrix cells.

### 5.2 Performance of Word Representations on Linguistic Tasks

**Linguistic Tasks and Datasets** We evaluated the word representations on four dataset, covering word similarity and relational analogy tasks. We used two datasets to evaluate pairwise word similarity: Finkelstein et al.'s *WordSim353* [13] and Bruni et al.'s *MEN* [4]. These datasets contain word pairs together with human-assigned similarity scores. The word vectors are evaluated by ranking the pairs according to their cosine similarities, and measuring the correlation (Spearman's $\rho$) with the human ratings.

The two *analogy* datasets present questions of the form "$a$ is to $a^*$ as $b$ is to $b^*$", where $b^*$ is hidden, and must be guessed from the entire vocabulary. The *Syntactic* dataset [22] contains 8000 morpho-

| WS353 (WORDSIM) [13] | | MEN (WORDSIM) [4] | | MIXED ANALOGIES [20] | | SYNT. ANALOGIES [22] | |
|---|---|---|---|---|---|---|---|
| Representation | Corr. | Representation | Corr. | Representation | Acc. | Representation | Acc. |
| SVD      (k=5)  | 0.691 | SVD      (k=1)  | 0.735 | SPPMI    (k=1)  | 0.655 | SGNS    (k=15) | 0.627 |
| SPPMI   (k=15) | 0.687 | SVD      (k=5)  | 0.734 | SPPMI    (k=5)  | 0.644 | SGNS    (k=5)  | 0.619 |
| SPPMI   (k=5)  | 0.670 | SPPMI   (k=5)  | 0.721 | SGNS    (k=15) | 0.619 | SGNS    (k=1)  | 0.59  |
| SGNS    (k=15) | 0.666 | SPPMI   (k=15) | 0.719 | SGNS    (k=5)  | 0.616 | SPPMI   (k=5)  | 0.466 |
| SVD      (k=15) | 0.661 | SGNS    (k=15) | 0.716 | SPPMI    (k=15) | 0.571 | SVD      (k=1)  | 0.448 |
| SVD      (k=1)  | 0.652 | SGNS    (k=5)  | 0.708 | SVD      (k=1)  | 0.567 | SPPMI   (k=1)  | 0.445 |
| SGNS    (k=5)  | 0.644 | SVD      (k=15) | 0.694 | SGNS    (k=1)  | 0.540 | SPPMI   (k=15) | 0.353 |
| SGNS    (k=1)  | 0.633 | SGNS    (k=1)  | 0.690 | SVD      (k=5)  | 0.472 | SVD      (k=5)  | 0.337 |
| SPPMI   (k=1)  | 0.605 | SPPMI   (k=1)  | 0.688 | SVD      (k=15) | 0.341 | SVD      (k=15) | 0.208 |

Table 2: A comparison of word representations on various linguistic tasks. The different representations were created by three algorithms (SPPMI, SVD, SGNS) with $d = 1000$ and different values of $k$.

syntactic analogy questions, such as "*good* is to *best* as *smart* is to *smartest*". The *Mixed* dataset [20] contains 19544 questions, about half of the same kind as in Syntactic, and another half of a more semantic nature, such as capital cities ("*Paris* is to *France* as *Tokyo* is to *Japan*"). After filtering questions involving out-of-vocabulary words, i.e. words that appeared in English Wikipedia less than 100 times, we remain with 7118 instances in Syntactic and 19258 instances in Mixed. The analogy questions are answered using Levy and Goldberg's similarity multiplication method [19], which is state-of-the-art in analogy recovery: $\arg\max_{b^* \in V_W \setminus \{a^*, b, a\}} \cos(b^*, a^*) \cdot \cos(b^*, b)/(\cos(b^*, a) + \varepsilon)$. The evaluation metric for the analogy questions is the percentage of questions for which the argmax result was the correct answer ($b^*$).

**Results** Table 2 shows the experiments' results. On the word similarity task, SPPMI yields better results than SGNS, and SVD improves even more. However, the difference between the top PMI-based method and the top SGNS configuration in each dataset is small, and it is reasonable to say that they perform on-par. It is also evident that different values of $k$ have a significant effect on all methods: SGNS generally works better with higher values of $k$, whereas SPPMI and SVD prefer lower values of $k$. This may be due to the fact that only positive values are retained, and high values of $k$ may cause too much loss of information. A similar observation was made for SGNS and SVD when observing how well they optimized the objective (Section 5.1). Nevertheless, tuning $k$ can significantly increase the performance of SPPMI over the traditional PPMI configuration ($k = 1$).

The analogies task shows different behavior. First, SVD does not perform as well as SGNS and SPPMI. More interestingly, in the syntactic analogies dataset, SGNS significantly outperforms the rest. This trend is even more pronounced when using the additive analogy recovery method [22] (not shown). Linguistically speaking, the syntactic analogies dataset is quite different from the rest, since it relies more on contextual information from common words such as determiners ("the", "each", "many") and auxiliary verbs ("will", "had") to solve correctly. We conjecture that SGNS performs better on this task because its training procedure gives more influence to frequent pairs, as opposed to SVD's objective, which gives the same weight to all matrix cells (see Section 3.2).

# 6   Conclusion

We analyzed the SGNS word embedding algorithms, and showed that it is implicitly factorizing the (shifted) word-context PMI matrix $M^{\text{PMI}} - \log k$ using per-observation stochastic gradient updates. We presented SPPMI, a modification of PPMI inspired by our theoretical findings. Indeed, using SPPMI can improve upon the traditional PPMI matrix. Though SPPMI provides a far better solution to SGNS's objective, it does not necessarily perform better than SGNS on linguistic tasks, as evident with syntactic analogies. We suspect that this may be related to SGNS down-weighting rare words, which PMI-based methods are known to exaggerate.

We also experimented with an alternative matrix factorization method, SVD. Although SVD was relatively poor at optimizing SGNS's objective, it performed slightly better than the other methods on word similarity datasets. However, SVD underperforms on the word-analogy task. One of the main differences between the SVD and SGNS is that SGNS performs *weighted* matrix factorization, which may be giving it an edge in the analogy task. As future work we suggest investigating weighted matrix factorizations of word-context matrices with PMI-based association metrics.

**Acknowledgements** This work was partially supported by the EC-funded project EXCITEMENT (FP7ICT-287923). We thank Ido Dagan and Peter Turney for their valuable insights.

## Footnotes

[1] In the algorithm described in [21], the negative contexts are sampled according to $p^{3/4}(c) = \frac{\#c^{3/4}}{Z}$ instead of the unigram distribution $\frac{\#c}{Z}$. Sampling according to $p^{3/4}$ indeed produces somewhat superior results on some of the semantic evaluation tasks. It is straight-forward to modify the PMI metric in a similar fashion by replacing the $p(c)$ term with $p^{3/4}(c)$, and doing so shows similar trends in the matrix-based methods as it does in `word2vec`'s stochastic gradient based training method. We do not explore this further in this paper, and report results using the unigram distribution.

[2]A notable exception is the case of syntactic similarity. For example, all verbs share a very strong negative association with being preceded by determiners, and past tense verbs have a very strong negative association to be preceded by "be" verbs and modals.

[3]The approach can be generalized to $W^{\text{SVD}_\alpha} = U_d \cdot (\Sigma_d)^\alpha$, making $\alpha$ a tunable parameter. This observation was previously made by Caron [7] and investigated in [6, 28], showing that different values of $\alpha$ indeed perform better than others for various tasks. In particular, setting $\alpha = 0$ performs well for many tasks. We do not explore tuning the $\alpha$ parameter in this work.

[4] `http://www.bitbucket.org/yoavgo/word2vecf`

[5] Since it is computationally expensive to calculate the exact objective, we approximated it. First, instead of enumerating every observed word-context pair in the corpus, we sampled 10 million such pairs, according to their prevalence. Second, instead of calculating the expectation term explicitly (as in equation 4), we sampled a negative example $\{(w, c_N)\}$ for each one of the 10 million "positive" examples, using the contexts' unigram distribution, as done by SGNS's optimization procedure (explained in Section 2).

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
