[Reviews · NeurIPS 2014]

Submitted by Assigned_Reviewer_7

This paper shows that the skip-gram model of Mikolov et al, when trained with their negative sampling approach can be understood as a weighted matrix factorization of a word-context matrix with cells weighted by point wise mutual information (PMI), which has long been empirically known to be a useful way of constructing word-context matrices for learning semantic representations of words. This is an important result since it provides a link between two (apparently) very different methods for constructing word embeddings that empirically performed well, but seemed on the surface to have nothing to do with each other. Using this insight, the authors then propose a new matrix construction and finds it performs very well on standard tasks.

The paper is mostly admirably clear (see below for a few suggestions on where citations could be added to make the relevant related work clear) and very nice contribution to have to explain what is going on in these neural language model embedding models.

The relationship between the global objective and the instance specific updates potential leave questions open about how exactly the word2vec objective relates to matrix factorization.

A few suggestions:

When you write “the well-known word-context PMI matrix from the word-similarity literature” give a citation (maybe Turney and Pantel, 2010?) or one of the other papers that does a big empirical comparison of the different association scores.

In the section “SGNS’s Objective”, say how this relates to noise contrastive estimation (NCE), see e.g.:
Mnih and Kavukcuoglu (2013). Learning word embeddings efficiently with noise-contrastive estimation. NIPS.

“data-driven unigram distribution” -> “empirical unigram distribution”

“partial derivative by x” -> “partial derivative with respect to x”

Section 3.2- can you construct the weighting matrix?

“Wikiepdia” -> “Wikipedia”
Summary: This is a clear, important paper that links matrix factorization of a particular kind of word-context matrix with the popular neural LM approach to learning word embeddings. Based on the insight, a new approach to creating a sparse word context matrix is proposed and found to be effective.

Submitted by Assigned_Reviewer_12

Lots of mileage has been made in NLP by looking at low rank
factorizations of say bigram matrices (or more generally, a word with
its embedded context). This can be justified by various hidden state
models. But it is often more natural to consider the log version of
the bigram matrix. So instead of factoring P_ij, something like
log(P_ij/P_i*P_j) is used. But, this introduces a whole can of worms
dealing with log(0) = -infinity. One usual way of dealing with this
is to simply replace the -infinities with zero. A better way is to use
the positive part:

max( log(P_ij/P_i*P_j) , 0 )

But, the zero is chosen completely arbitrarily. This paper truncates
at log(k) instead of at zero:

max( log(P_ij/P_i*P_j) - log(k) , 0 ).

They call this the Shifted PPMI.

The fact that it takes 4 pages before the shifted PPMI is introduced
in equation (11) is fine by me since the material covered is nicely
presented and well argued.

They then try out their new method on a variety of NLP tasks. They do
acceptably well on these tasks. Since they are all pretty carefully
mined already, I don't expect anything to do wonderfully better than
the existing methods.
Summary: A new modification to an existing NLP methodology is introduced. It is argued to be natural, and found to perform nicely on NLP tasks.

Submitted by Assigned_Reviewer_36

The paper investigates the skip-gram model of Mikolov et al., more specifically, the more successful 'negative sampling' version. The submission draws a connection between the word2vec objective and a form of PMI (pointwise mutual information) measure, and, guided by these analyses, proposes a simple modification of PMI (shifting it by a constant, and replacing negative values with zero).

The paper is generally well-written and interesting in many aspects, however I have a few reservations:

1) The connection between the skip gram model and factorizing log P(w | c ) / P(w) is unsurprising, especially for the soft-max version of the model (also recall the fairly established practice of applying LSA in log space). Still, it is nice to see a more formal look into this connection for the negative-sampling version.

2) The authors may miss another role of the constant k in the context of word2vec (and perhaps more important one?). Unlike, their theoretical analyses, word2vec does not perform marginalization over the entire vocabulary and multiplication of this expectation by k, but rather draws k samples. Thus, another role of k is to produce a more reasonable approximation of the corresponding expectation (~ the softmax partition function). This may be especially important given that word2vec is normally run only for a very small number of iterations over the training set. It may be interesting to see if what matters is the 'shift' role of k (as argued by the authors) or the approximation / exploration role of k. This is actually fairly easy to do by introducing 2 constants (m and l) instead of k. Instead of considering k negative samples and summing over them, one could consider m samples, normalize them (i.e. divide the sum by m) and multiply the result by l. This would help to disentangle the 2 roles: increasing m would improve the quality of the approximation (without affecting the shift), and increasing l would increase the 'shift' (without affecting the quality of the approximation).

3) Overall, this "shifting-by-log k" extension seems a little ad-hoc (and not so helpful in practice). I am not sure I can really buy its motivation — it seems to be motivated by the fact that the negative sampling objective is ‘using’ the ‘shift’. But it is not really clear that this is really desirable, and it is not clear whether this is really that useful (see the point (2) above), especially in the context of PMI. On the one hand it results in ignoring low ‘correlations’ but on the other it is doing more than that as the cosine similarity is not shift invariant. Some more discussion / analyses would be helpful.

24 The authors look only into the local objective (i.e. partial derivative of the objective on a specific word-context pair). It may make more sense to look into the global objective, as this objective explicitly incorporates weighting for the individual terms in the factorization problem. This is mentioned in section 3.2 but I was surprised that it has not been done in 3.1.

5) I am really not sure what these small values for SPPMI in Table 1 are telling to us. SPPMI is different from the objective only for negative terms, so this is probably where the difference is coming from. Is this delta so small because negative terms are infrequent? I am slightly unsure if the results are also affected by the quality of the objective approximation you are using (here I refer to the approximations used to compute the objective, as described in section 5.1). Some additional discussion may be helpful.

6) Overall, section 5.1 is somewhat questionable as SPPMI does not really optimize the objective as no embeddings are really induced. For any objective function, optimizing terms independently (i.e. ignoring any interaction between them) would result in a better 'solution'. Not really sure what this tells us though.

7) Though it is interesting to see that using PPMI results in comparable or better results on many tasks, this is not a new result as similar observations were made in Levy and Goldberg (CoNLL 2014, cited).

Summary: A well-written paper which can potentially stimulate some interesting discussion. However, the theoretical results are not really ground breaking and the empirical results are not very exciting.
Author Feedback
Author rebuttal: Thank you for your thoughtful reviews.

We share Reviewer 3's opinion that the main contribution of the paper is that "it provides a link between two (apparently) very different methods for constructing word embeddings that empirically performed well, but seemed on the surface to have nothing to do with each other." In particular, we draw a connection between co-occurrence matrix-based methods and log-bilinear neural-inspired models such as SGNS and NCE. Given the current interest in neural-network inspired word representations, and specifically the rising popularity of word2vec, we feel that it is important to understand what these algorithms are doing, and to be able to properly compare and relate them to other methods of constructing word representations.

The second contribution, shifted-PPMI, is indeed incremental, and is not necessarily meant to be used as the new state-of-the-art. Instead, shifted-PPMI is presented to allow a more exact empirical comparison between different methods of factorization (none, SVD, SGNS). There are several parameters in word2vec (such as the number of negative samples k) that can be equally applied to PMI matrix-based representations. If these parameters are not accounted for, the empirical comparison between traditional matrix-based and word2vec-style methods may lead to erroneous conclusions.

WRT to the empirical results, Reviewer 2 pointed out (point 7) that getting to the SPPMI models to perform as well as or better than word2vec is not surprising, as this has been demonstrated before by [Levy and Goldberg, 2014]. In fact, the results in Levy and Goldberg are only on analogy-type tasks. For word-similarity tasks, the myriad of experiments in [Baroni et al, 2014] suggest that word2vec embeddings vastly outperform the matrix-based representations. We believe our paper is the first to demonstrate a matrix-based model outperforming word2vec on a word similarity task, and, as can be seen in table 2, shifting-by-k is instrumental in this result.

WRT Reviewer 2's comment (2) on the role of k: indeed, the parameter was probably intended to improve the approximation. However, the mathematical analysis reveals that it also acts as a regularizer for the ratio between positive and negative examples. Indeed, we performed experiments (not reported in the paper) in which we ran word2vec's optimization for several epochs over the training corpus, while fixing k to 1. The effect of the number of epochs on the resulting model's accuracy was negligible when compared to the effect of setting different k values for the negative sampling.

WRT Reviewer 3's comment on NCE: after submitting the paper, we performed a similar analysis of Mnih et al’s model, and found that it too performs implicit matrix factorization of shifted log-conditional probabilities. Specifically:
w*c = log(#(w,c) / #(c)) + log(k)
We will mention this in the final version.

WRT other comments, we welcome your suggestions and will modify the final version accordingly.

[Omer Levy and Yoav Goldberg. "Linguistic Regularities in Sparse and Explicit Word Representations". CoNLL 2014.]

[Marco Baroni, Georgiana Dinu, and German Kruszewski. "Don't Count, Predict! A Systematic Comparison of Context-Counting vs. Context-Predicting Semantic Vectors". ACL 2014.]